# Exploring the overlap between rheumatoid arthritis susceptibility loci and long non-coding RNA annotations

**James Ding**[1]*, **Chenfu Shi**[1], **John Bowes**[1,2], **Stephen Eyre**[1,2], **Gisela Orozco**[1,2]

**1** Division of Musculoskeletal and Dermatological Sciences, Centre for Genetics and Genomics Versus Arthritis, School of Biological Sciences, Faculty of Biology, Medicine and Health, The University of Manchester, Manchester, England, United Kingdom, **2** NIHR Manchester Biomedical Research Centre, Manchester Academic Health Science Centre, Manchester University NHS Foundation Trust, Manchester, England, United Kingdom

* james.ding@manchester.ac.uk

**Data Availability Statement:** The datasets analysed during the current study are available from FANTOM, GWAS Catalogue, LNCipedia, MiTranscriptome, Roadmap Epigenomics project, and Sequenced Read Archive data repositories:

## Abstract

Whilst susceptibility variants for many complex diseases, such as rheumatoid arthritis (RA), have been well characterised, the mechanism by which risk is mediated is still unclear for many loci. This is especially true for the majority of variants that do not affect protein-coding regions. lncRNA represent a group of molecules that have been shown to be enriched amongst variants associated with RA and other complex diseases, compared to random variants. In order to establish to what degree direct disruption of lncRNA may represent a potential mechanism for mediating RA susceptibility, we chose to further explore this overlap. By testing the ability of annotated features to improve a model of disease susceptibility, we were able to demonstrate a local enrichment of enhancers from immune-relevant cell types amongst RA susceptibility variants ($\log_2$ enrichment 3.40). This was not possible for lncRNA annotations in general, however a small, but significant enrichment was observed for immune-enriched lncRNA ($\log_2$ enrichment 0.867002). This enrichment was no longer apparent when the model was conditioned on immune-relevant enhancers ($\log_2$ enrichment -0.372734), suggesting that direct disruption of lncRNA sequence, independent of enhancer disruption, does not represent a major mechanism by which susceptibility to complex diseases is mediated. Furthermore, we demonstrated that, in keeping with general lncRNA characteristics, immune-enriched lncRNA are expressed at low levels that may not be amenable to functional characterisation.

## Introduction

In keeping with other complex diseases, rheumatoid arthritis (RA) susceptibility loci are mainly non-coding, with relatively few variants having a potential impact upon the coding sequence for a protein [1,2]. Enhancers have been identified as likely to mediate disease susceptibility at many loci. Evidence to support this generalization includes demonstrated effects of genome wide association study (GWAS) variants on enhancers at individual loci [3], as well as an enrichment of RA GWAS variants amongst enhancers from relevant cell types [4].

http://fantom.gsc.riken.jp/5/suppl/Hon_et_al_2016/data/, ftp://ftp.ebi.ac.uk/pub/databases/gwas/summary_statistics/OkadaY_24390342_GCST002318, https://lncipedia.org/download, http://mitranscriptome.org/download/, https://egg2.wustl.edu/roadmap/data/byFileType/chromhmmSegmentations/ChmmModels/core_K27ac/jointModel/final/, and https://www.ncbi.nlm.nih.gov/sra, respectively.

**Funding:** This research was funded by the Wellcome Trust and Versus Arthritis and supported by the NIHR Manchester Biomedical Research Centre. JD, CS and GO are funded by the Wellcome Trust (JD and GO, 207491/Z/17/Z; CS, 215207/Z/19/Z) and by Versus Arthritis (21754 and 21348). JB and SE are funded by Versus Arthritis (JB, 21754; SE, 21754 and 21348). The funders had no role in study design, data collection and analysis, decision to publish, or preparation of the manuscript. There was no additional external funding received for this study.

**Competing interests:** The authors have declared that no competing interests exist.

Alternative non-coding elements, such as long non-coding RNA (lncRNA) may also play a role in mediating the increased risk associated with non-coding variants.

lncRNA are a heterogeneous class of molecules that are defined based on a lack of protein-coding potential and a minimum transcribed length of 200 nucleotides. Whilst discrete subcategories exist, such as long intergenic non-coding RNAs, promoter associated-lncRNAs or antisense lncRNA with some discriminatory characteristics, including genomic context, overlapping chromatin marks, length and structure [5], it can still be difficult to discriminate a genuine lncRNA annotation from a product of spurious transcription. Many individual lncRNA have been functionally characterised, with gene regulation featuring frequently amongst the wide variety of roles described. One important subcategory was identified following observations of RNA polymerase II recruitment and transcription at enhancers [6]. Often described as enhancer derived RNA (eRNA), expression of these transcripts is highly correlated with enhancer activity [7] and increasing evidence suggests that these ncRNA may contribute towards enhancer function, although the precise mechanism is still unclear [8].

There is some evidence to suggest that GWAS susceptibility variants are enriched amongst lncRNA [9,10]. However, using conventional methods of determining whether annotations overlap more than can be expected by chance it is difficult to appropriately account for confounding factors, such as chromosomal compartments or chromatin accessibility. Using the *ab initio* MiTranscriptome assembly (58,648 lncRNA), which was generated using a large collection of RNA sequencing libraries, GWAS SNPs were demonstrated to be enriched in lncRNA, compared to other SNPs tested for in GWAS analyses [9]. This enrichment was also observed using either GWAS SNPs or probabilistically identified causal SNPs (PICS) and lncRNA from the functional annotation of the human genome (FANTOM) cap-analysis gene expression (CAGE) associated transcriptome (CAT) assembly (27,919 lncRNA), generated using CAGE datasets in combination with existing assemblies [10]. Using the tissue specific nature of the FANTOM CAT annotation it was also possible to demonstrate that this enrichment was markedly higher when testing specifically for an enrichment of immune-relevant GWAS PICS in immune-expressed lncRNA transcripts.

Despite these studies, the relevance of lncRNA to the study of individual complex diseases, such as RA, remains unclear. This is especially true given the overlap in genomic locality and function between enhancers and lncRNA. We chose to investigate the overlap between lncRNA annotations, enhancer annotations and GWAS SNPs associated with RA susceptibility (Fig 1), with the aim of establishing to what degree the direct disruption of lncRNA by RA-associated variants may contribute to the mediation of disease risk. Central to our investigation is the use of the fgwas algorithm [11], which tests the ability of individual annotations to improve a probabilistic model of disease susceptibility, constructed using GWAS summary statistics. Using this method a local enrichment is estimated, that takes into consideration the non-random distribution of potentially confounding genomic features. In addition, it is possible to model multiple traits and establish the degree to which they are independently predictive.

## Materials and methods

### Enrichment testing

Enrichment testing was performed using RA summary statistics [1] and fgwas v0.3.6 [11], with the "-cond" option called for conditional analyses. Enrichment estimates are reported as outputted by fgwas, on a $\log_2$ scale, such that positive values indicate an enrichment and negative values indicate a depletion of the given annotation amongst disease susceptibility variants. Chromatin state data was obtained from the Roadmap Epigenomics project [12], with the

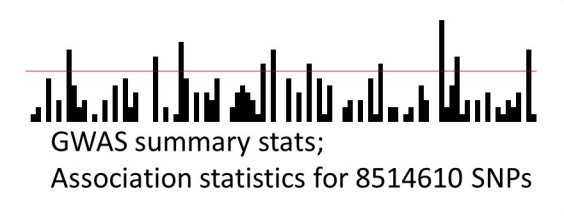

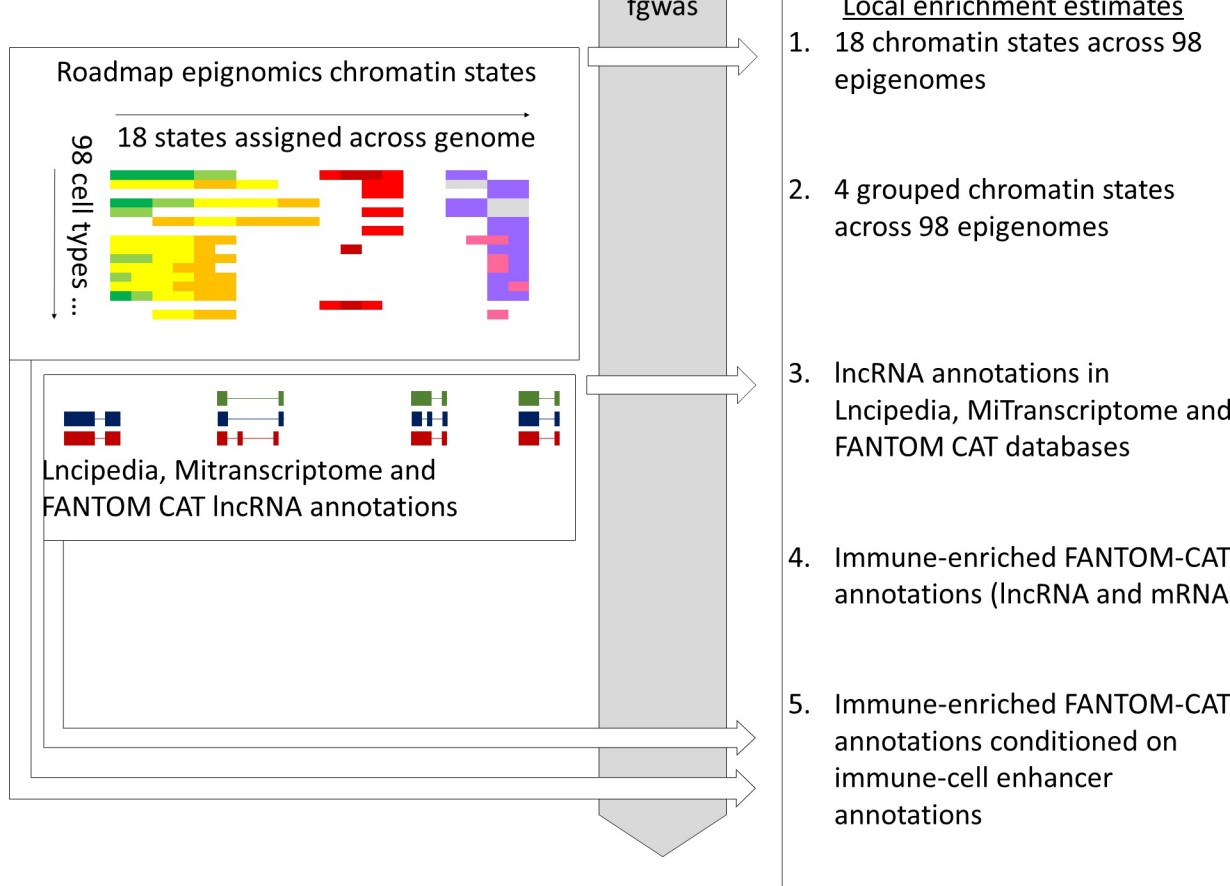

**Fig 1. Visual interpretation of analysis pipeline.** GWAS summary stats were used to inform a probabilistic model of RA susceptibility. Various features were taken from publically available data and their ability to improve this model was tested, features that improve the model can be thought of as enriched amongst RA susceptibility variants.

expanded, 18-state model used for all 98 corresponding epigenomes. These chromatin states originate from a hidden Markov model, based on H3K4me3, H3K4me1, H3K36me3, H3K27me3, H3K9me3 and H3K27ac occupancy data. In our analyses, the 18-states were combined to form four exclusive annotations as outlined in Table 1.

The following lncRNA datasets were interrogated: Lncipedia v5.2 [13], miTranscriptome v2 [9] and FANTOM CAT (robust) [10].Immune-relevant enhancers were defined as genomic regions annotated as enhancers in cell-types originating from "Blood and T cell" or "HSC and B cell" anatomical locations by the Roadmap Epigenomics project [12]. These include a total

**Table 1. Consolidation of Roadmap epigenomics 18-state chromatin states into four exclusive annotations.**

| TSSs | Transcription | Enhancers | Repressed Chromatin |
|---|---|---|---|
| 1. Active TSS | 5. Strong transcription | 7. Genic enhancer 1 | 12. ZNF genes and repeats |
| 2. Flanking TSS | 6. Weak Transcription | 8. Genic enhancer 2 | 13. Heterochromatin |
| 3. Flanking TSS upstream | | 9. Active enhancer 1 | 16. Repressed polycomb |
| 4. Flanking TSS downstream | | 10. Active enhancer 2 | 17. Weak repressed polycomb |
| 14. Bivalent/poised TSS | | 11. Weak enhancer | 18. Quiescent/low |
| | | 15. Bivalent enhancer | |

TSS, transcription start site; ZNF, zinc finger. The chromatin state number employed by the Roadmap Epigenomics Project is included to enable easy comparison.

of 17 partially overlapping primary cell populations, isolated from peripheral and umbilical cord blood; including peripheral blood mononuclear cells, B cells, Natural killer cells, haematopoietic stem cells, and various subtypes of T cells. The definition of Immune-enriched lncRNA is based on the underlying sample ontology and was wholly adopted from Hon et al. requiring: detection in at least 50% of immune-relevant samples, 5 x higher expression in immune-relevant samples than in other samples and $P < 0.05$ in a one-tailed Mann–Whitney rank sum test [10].

## Expression profiling

Raw RNA-seq reads were downloaded from the Roadmap Epigenomics project for primary T-helper cells (Roadmap Epigenomics epigenome ID E043, SRA accession SRR644513 and SRR643766, originating from a 37 y old Hispanic male, and 21 y old Caucasian male, respectively) [12]. Reads were then filtered for quality, adapter content and polyA tails using fastp version 19.7, with default settings and polyX tail trimming enabled. Transcripts were quantified using Salmon version 13.1 [14] using suggested settings ("quant" mode and "–validate-mappings") and using the reference index generated from the FANTOM CAT robust database [10]. Transcripts quantifications (reported as Transcripts per million, TPM) were then remapped to genes and summed for each gene. CAGE transcript counts per million (CPM) were taken from FANTOM CAT for T-helper cells (CL_0000084_T_cell), as published [10].

Statistical difference between the distributions of immune-enriched lncRNA and mRNA abundance was established using a two-sided Z-test, with no assumption of equal variance.

## Results

### Enhancer annotations from immune-relevant cell types are enriched amongst RA susceptibility variants

An enrichment of RA PICS has previously been demonstrated amongst *cis*-regulatory elements that are active in T-helper cells and lymphoblastoid cells [4], using data from the high density Immunochip custom SNP array. In order to establish confidence in the ability of fgwas to identify similar enrichments we sought to validate this evidence of enrichment using a more inclusive approach that incorporates the probability of association for all imputed SNPs taken from the most recent European RA GWAS meta-analysis (8 514 610, p value ranges from 1.0 to $1.94 \times 10^{-280}$) [1]. In order to achieve this, chromatin state data taken from the Roadmap Epigenomics project [12] was incorporated in a model of RA susceptibility. Using enrichment estimates generated using the expanded 18-state model, it is possible to discern an enrichment of certain chromatin states, such as genic enhancers, active enhancers and weak enhancers in immune-relevant cell types, such as B cells, T cells and monocytes (panel A in S1 Fig), however the confidence intervals (CIs) for estimates are broad for many states, likely due to a reduced

abundance of these annotations (panel B and C in S1 Fig). The size of CIs was improved by combining states to generate four more easily interpretable annotations (enhancers, transcription start sites (TSSs), transcription, and repressed chromatin; Fig 2). In keeping with our understanding of RA, the highest level of enrichment was observed for enhancer annotations in immune-relevant cell types, with regulatory T cells showing the highest enrichment ($\log_2$ enrichment 3.17, 95% CI 2.58; 3.75015). In immune-relevant cell types TSSs were also enriched, whilst repressed chromatin was depleted (Fig 2).

## lncRNA annotations show negligible enrichment amongst RA susceptibility variants

We applied fgwas to test for an enrichment of lncRNA amongst RA susceptibility variants, using lncipedia, a large lncRNA database curated from a number of sources [13], the MiTranscriptome assembly [9], and the FANTOM CAT assembly [10]. Using fgwas, MiTranscriptome lncRNA genes show a level of depletion amongst RA susceptibility variants comparable to repressed chromatin. MiTranscriptome lncRNA exons and both genes and exons from either FANTOM CAT or lncipedia all show negligible enrichment (Fig 3A).

## lncRNA annotations with enriched expression in immune-relevant cells are subtly enriched amongst RA susceptibility variants

Uniquely, FANTOM CAT transcripts are associated with tissue specific expression data. As in the original FANTOM CAT publication, we took advantage of this additional information, to test for an enrichment of lncRNA whose expression is enriched in immune-relevant cell types amongst RA susceptibility loci. This approach demonstrated a subtle enrichment of lncRNA genes ($\log_2$ enrichment 0.867, 95% CI 0.0554; 1.57) and similar level of enrichment for their exons, albeit with an increased confidence interval ($\log_2$ enrichment 0.799, 95% CI 2.30; 1.94). FANTOM CAT mRNA annotations, whose expression is enriched in immune-relevant cell types, were included in order to provide a comparison. Genic mRNA annotations showed a similar level of enrichment as lncRNA genes, with mRNA exons exhibiting slightly higher enrichment (Fig 3B).

## The subtle enrichment of immune-enriched lncRNA observed is not independent of immune-relevant enhancer annotations

Given the strong enrichment of immune-relevant enhancers amongst RA susceptibility loci and the established overlap between lncRNA annotations and enhancers, we were interested to investigate the independence of these variables using fgwas. In this conditional analysis, a residual enrichment of immune-enriched FANTOM CAT annotations was tested after the enrichment of immune-relevant enhancers ($\log_2$ enrichment 3.40, 95% CI 2.54; 4.58) was accounted for (Fig 4). Interestingly, both lncRNA and mRNA annotations no longer show significant enrichment, indicating that once enrichment of susceptibility variants in enhancers has been accounted for no remaining enrichment of mRNA or lncRNA is apparent.

## Immune-enriched lncRNA are expressed at low levels in RA relevant cell types

lncRNA are generally considered to exhibit low expression levels and high tissue specificity that can make them difficult to study using conventional methods. Given their enrichment for RA susceptibility variants we were interested to establish whether this description applied to FANTOM CAT immune-enriched lncRNA in an RA relevant cell type. Using randomly

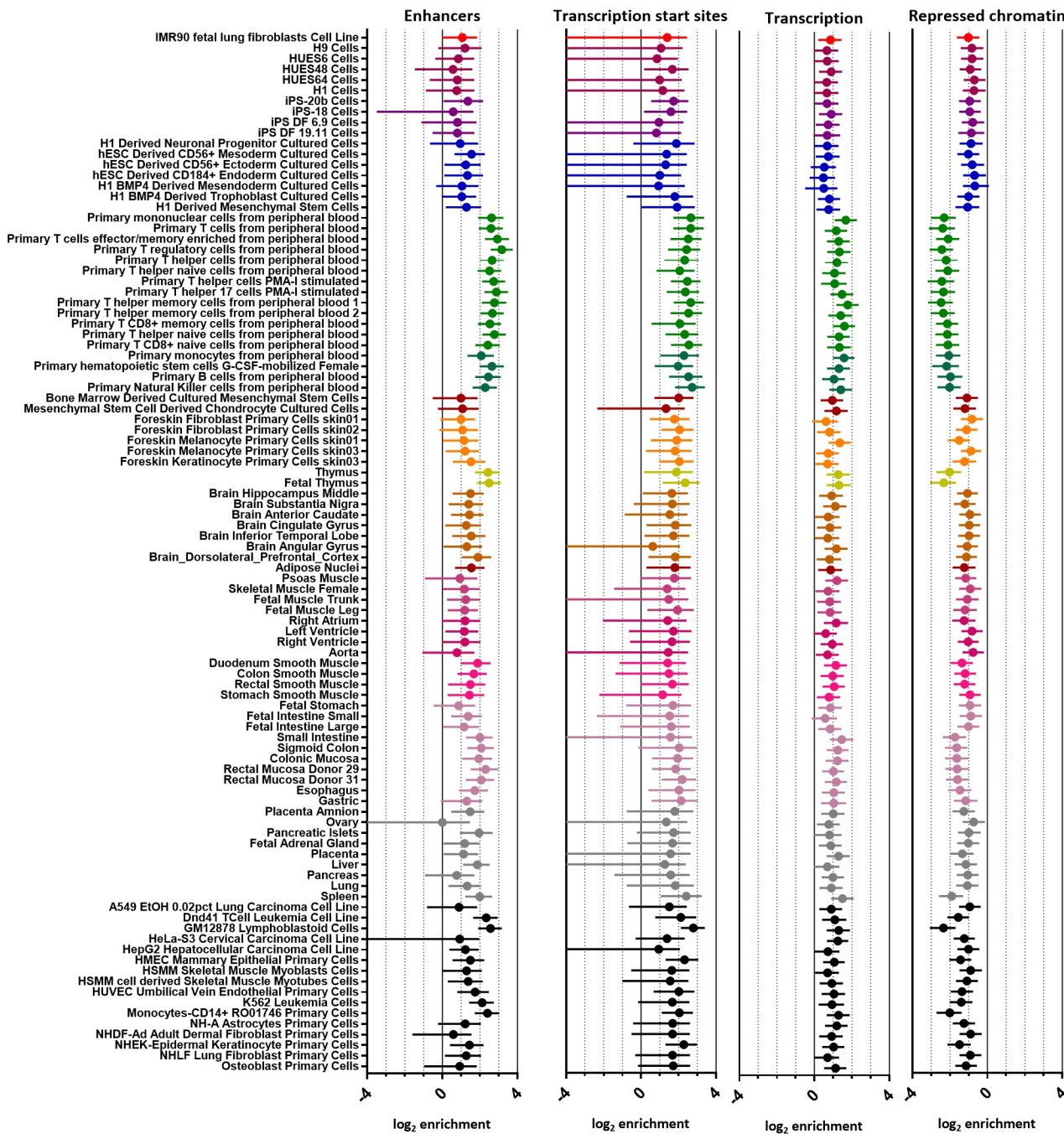

**Fig 2. Enrichment of chromatin state groups amongst RA susceptibility variants for 98 cell types.** Estimates for enrichment of combined chromatin state groupings are illustrated for all 98 cell types annotated within the Roadmap Epigenomics 18-state model. Cell-types are ordered and coloured according to the clustering established by the Roadmap Epigenomics project, with immune-relevant cell types coloured green. Estimates and confidence intervals are clipped at axis limits, where applicable.

primed Roadmap Epigenomics RNA-seq data from primary T-helper cells, the distribution of expression levels for FANTOM CAT immune-enriched lncRNA is significantly lower than that of FANTOM CAT immune-enriched mRNA (p = 1.04 x $10^{-34}$, Fig 5A median lncRNA transcripts per million reads (TPM); 0.257, vs 124 for mRNA). The same is true using less-conventional expression profiling methods, such as those employed by FANTOM CAT, which

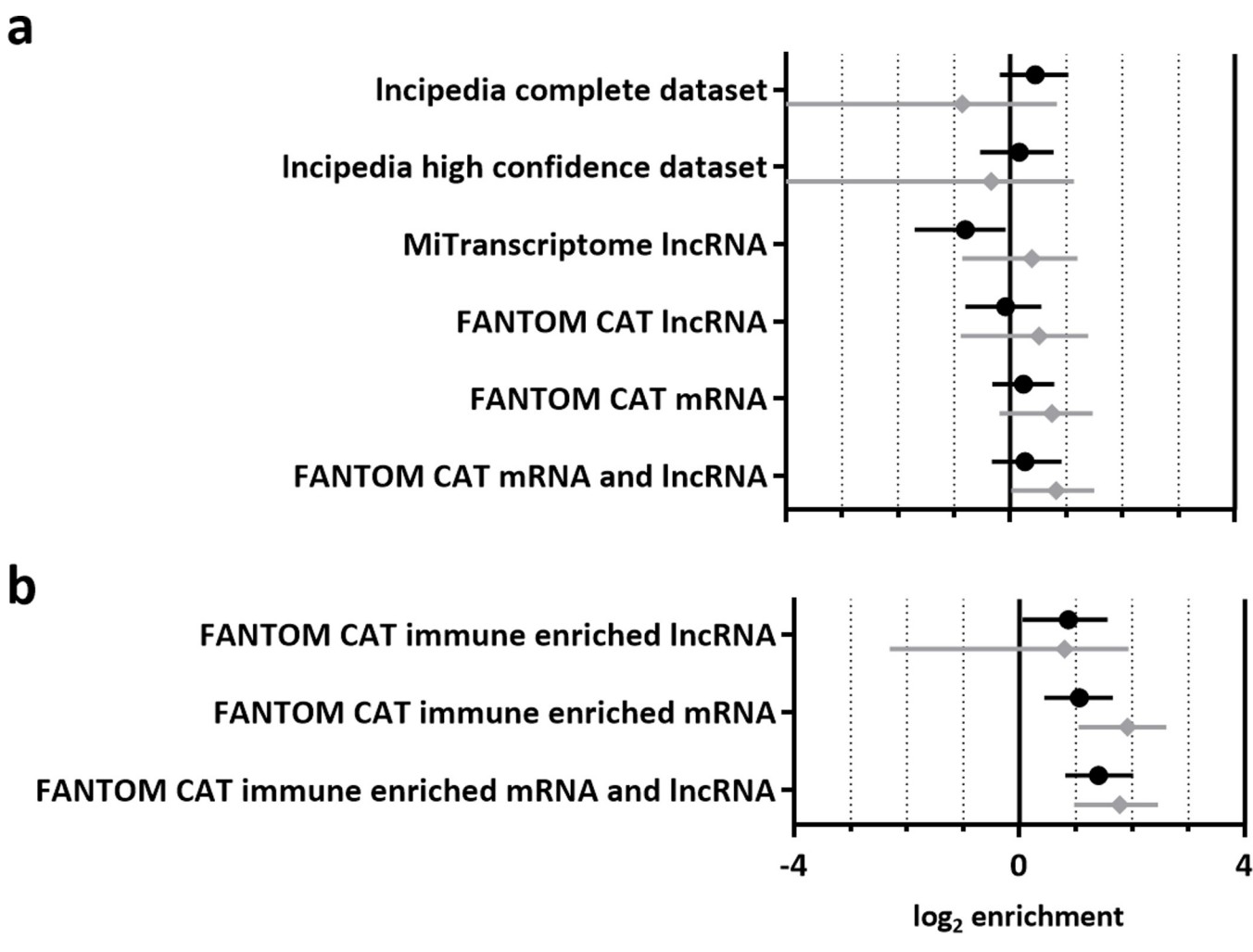

**Fig 3. Enrichment of lncRNA annotations amongst RA susceptibility variants.** Estimates for the enrichment of genic (black circle) and exonic (grey diamond) annotations from a variety of lncRNA containing databases, including 95% confidence intervals (A). Separate estimates are included for annotations identified as exhibiting enriched expression in immune-relevant cells (B).

offer improved sensitivity for the detection of transcripts of low abundance (p = 1.01 x 10$^{-24}$, Fig 5B median lncRNA counts per million reads (CPM); 1.43, vs 71.7 for mRNA). 90% of immune-enriched lncRNA have abundance lower than 85.2% of immune-enriched mRNA in Roadmap Epigenomics RNA-seq data or 75.6% of immune-enriched mRNA in FANTOM CAT CAGE data.

## Discussion

By incorporating both cell-type specific enhancer and lncRNA annotations into a probabilistic model of RA susceptibility it is possible to demonstrate their respective levels of enrichment amongst RA susceptibility variants. Whilst previous studies have demonstrated an enrichment of lncRNA compared with randomly shuffled annotations these analyses fail to take into consideration the complex organisation of the genome and are easily confounded by alternative features. In our analysis, which incorporated lncRNA from various databases, it was only possible to demonstrate a subtle enrichment of lncRNA whose expression was previously identified as being enriched in relevant cell-types.

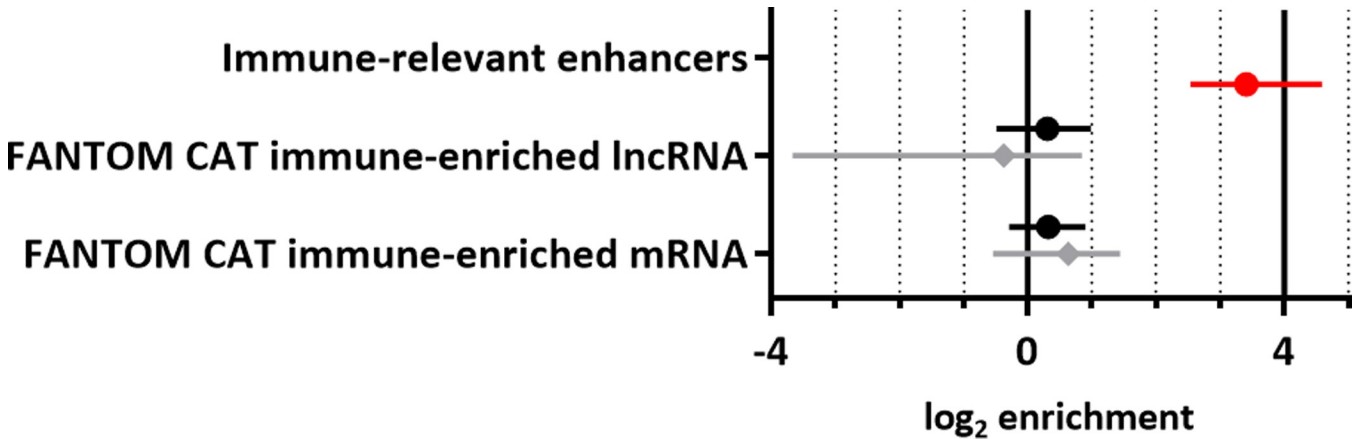

**Fig 4. Enrichment of immune-enriched lncRNA amongst RA susceptibility variants after conditioning on chromatin state data.** The influence of immune-relevant enhancers (red circle) was fixed in a probabilistic model of RA susceptibility to determine whether the subtle enrichment of FANTOM CAT immune-enriched lncRNA or mRNA adds any additional predictive information and is therefore independently enriched. Genic (black circle) and exonic (grey diamond) annotations were both tested. As may be expected given the magnitude of enrichments observed, after accounting for the effect of immune-enriched FANTOM CAT annotations, the residual enrichment of immune-relevant enhancers is not dramatically reduced (S2 Fig).

By conditioning a model of RA susceptibility on enhancer annotations from immune-relevant cell types it is possible to test the independence of additional features. This demonstrated that the subtle enrichment observed for immune-relevant lncRNA is entirely explained by immune-relevant enhancer annotations. Interestingly, the same is true of mRNA annotations, indicating that in both instances the primary influence of susceptibility variants is in affecting non-coding regulatory elements, with any effect on mRNA and lncRNA being secondary and/or indirect. This suggests that the majority of genetic variance in disease susceptibility is mediated through disruption of regulatory elements and not through direct disruption of transcript sequences. This observation is in keeping with those made previously, relating to the minimal overlap between the coding sequence of mRNA and RA susceptibility variants. It is also consistent with the existence of well-characterised effects of RA susceptibility variants on coding

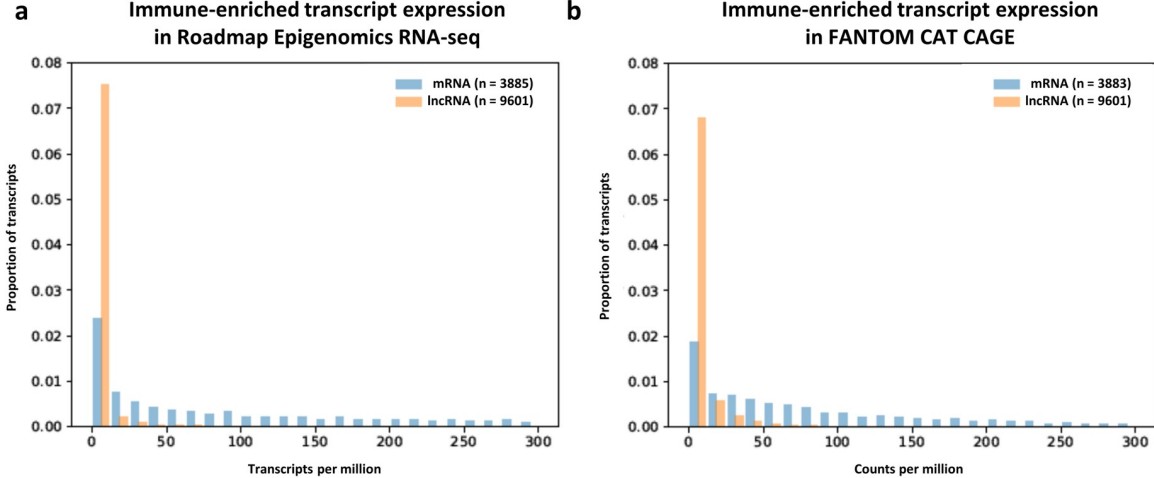

**Fig 5. Distribution of immune-enriched lncRNA and mRNA expression levels in primary T-helper cells.** Staggered bars are used to illustrate the proportion of transcripts whose expression falls in bins of 25 million transcripts, or counts, in Roadmap Epigenomics RNA-seq data (A) and FANTOM CAT CAGE data (B), respectively.

regions, such as for the HLA proteins [15], which convey a significant, but not exhaustive proportion of risk.

Similarly, this analysis does not rule out the relevance of lncRNA at individual loci, in fact it is worth noting that there is evidence to suggest that C5T1lncRNA may mediate risk at a RA risk locus located at chromosomal position 9q33.2, with RA associated variants falling within a C5T1lncRNA exonic region [16]. Our analysis, however, precludes a general effect of RA susceptibility variants on the transcribed sequence of lncRNA independent of enhancer disruption. Sequence-specific functions of lncRNA are, therefore, unlikely to mediate a significant proportion of the risk-modifying effect associated with RA susceptibility variants. Furthermore, the distribution of FANTOM-CAT immune-enriched lncRNA expression levels in primary T-helper cells highlight the difficulties associated with studying lncRNA, whose expression is typically very low and highly cell-type specific. It should however be highlighted that whilst these obstacles may complicate the functional characterisation of lncRNA, low, highly tissue specific expression on its own does not preclude a biologically significant role.

In our analysis, we have sought to highlight the difficulties associated with characterising a role for lncRNA that overlap disease susceptibility variants in mediating disease susceptibility. These difficulties have also undoubtedly contributed to limitations in the accuracy and completeness of the lncRNA annotations used, where lncRNA with low expression or with expression levels that are highly specific to a certain developmental stage, condition, or cell type are likely to be underrepresented.

Many mechanisms have been described, whereby lncRNA may impact disease susceptibility irrespective of disruptions to their DNA sequence and our findings do not limit the likelihood that these mechanisms may be important in RA susceptibility. For example, transcription of a lncRNA at a particular locus may reinforce a permissive chromatin state that is required for accurate regulation of another gene. However, where such a mechanism is found to be important in mediating the effects of a disease susceptibility variant, it is perhaps most likely that the primary impact of the disease susceptibility variants is in disrupting a regulatory element that controls the lncRNA expression. In such instances, it is inherently difficult to disentangle the functional contribution of the lncRNA and regulatory element.

The analyses performed are specific to RA, however it is assumed that similar results would be reached using GWAS data, enhancer annotations and lncRNA annotations relevant to other diseases. This would, therefore, suggest that the enrichment of lncRNA annotations amongst GWAS variants observed by others [9,10], may result from a high degree of overlap between regulatory features such as enhancers and lncRNA and other confounding features, such as active and inactive genomic compartments. As a generalisation, when it comes to functional characterization of variants associated with complex genetic disorders, sequence-specific lncRNA functions is unlikely to represent an attractive area for study; lncRNA are not independently enriched amongst such variants and are difficult to study, due to low expression levels. Despite these results, dysregulation of lncRNA expression could still play a role in RA and similar diseases, with disease associated variants affecting regulatory elements, such as enhancers that control lncRNA expression.

fgwas represents a useful tool for studying the enrichment of different features amongst susceptibility variants, especially when used in combination with Roadmap Epigenomics data in order to identify cell-types and tissues that are relevant for disease susceptibility. Whilst using this tool we observed that the number and size of annotations that are tested have a strong influence, both on the confidence with which any enrichment is estimated, as well as on the extent of that enrichment. It is likely that this may explain some of the subtle differences observed between different lncRNA databases.

Our analyses highlight the caveats associated with inferring functional relevance for a given feature, based purely on the observation of enrichment over a genomic background, as well as the care that must be taken when attempting to interpret such enrichments. By deriving enhancer and lncRNA annotations from entirely different sources we have tried to ensure that the demonstrated dependence is not self-fulfilling, as it may have been if we defined immune-relevant lncRNA based on underlying chromatin states.

In conclusion, using fgwas and Roadmap Epigenomics chromatin state data it is possible to identify cell types and chromatin states of relevance to complex diseases, such as RA. In the case of RA this is predominantly enhancers and transcription start sites from immune-relevant cell types. It is also possible to test the association of alternative features and establish their independence from chromatin states. Here, a previously described enrichment of lncRNA amongst GWAS susceptibility loci was explored for RA. Immune-enriched lncRNA from the FANTOM-CAT database were found to be enriched amongst RA susceptibility loci, however, this enrichment was not apparent when chromatin-state data was taken into account.

Our results suggest that regulatory elements, such as enhancers, are likely to mediate the vast majority of variance in risk associated with RA and other complex diseases, with no substantial independent contribution being made by direct disruption of lncRNA sequences. Because of this, and the difficulties associated with detecting transcripts of such low abundance, sequence-specific lncRNA function does not represent the most attractive area for study with respect to RA susceptibility, except in the case of in depth characterisation of individual loci.

## Supporting information

**S1 Fig. Enrichment of chromatin state annotations amongst RA susceptibility variants.** Estimates for enrichment of individual states are illustrated for 98 cell types using the Roadmap Epigenomics 18-state model (a). Similar states were grouped into four groups for all immune-relevant primary cell-types (b), as individual states often gave very broad 95% confidence intervals (c). Cell-types are ordered according to the clustering established by the Roadmap Epigenomics project, with chromatin states reordered according to their subsequent grouping. Estimates and confidence intervals are clipped at axis limits, where applicable. (TIF)

**S2 Fig. Enrichment of immune-relevant enhancers amongst RA susceptibility variants after conditioning on immune-enriched transcripts.** The influence of FANTOM CAT immune-enriched lncRNA and mRNA was fixed in a probabilistic model of RA susceptibility to confirm the independent enrichment of immune-relevant enhancer chromatin states. (TIF)

## Acknowledgments

The authors would like to acknowledge the assistance given by IT Services and the use of the Computational Shared Facility at The University of Manchester.

## Author Contributions

**Conceptualization:** James Ding, Chenfu Shi, John Bowes, Gisela Orozco.

**Funding acquisition:** Stephen Eyre, Gisela Orozco.

**Investigation:** James Ding, Chenfu Shi.

**Methodology:** Chenfu Shi, John Bowes.

**Supervision:** John Bowes, Stephen Eyre, Gisela Orozco.

**Visualization:** James Ding, Chenfu Shi.

**Writing – original draft:** James Ding.

**Writing – review & editing:** James Ding, Chenfu Shi, John Bowes, Stephen Eyre, Gisela Orozco.

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
