## [Decision Letter · Decision Letter 0]

17 Dec 2019

PONE-D-19-27368

Exploring the overlap between rheumatoid arthritis susceptibility loci and long non-coding RNA annotations

PLOS ONE

Dear Dr Ding,

Thank you for submitting your manuscript to PLOS ONE. After careful consideration, we feel that it has merit but does not fully meet PLOS ONE’s publication criteria as it currently stands. Therefore, we invite you to submit a revised version of the manuscript that addresses the points raised during the review process.

Please, implement suggestions, I do not see major problems.

We would appreciate receiving your revised manuscript by Jan 31 2020 11:59PM. To enhance the reproducibility of your results, we recommend that if applicable you deposit your laboratory protocols in protocols.io, where a protocol can be assigned its own identifier (DOI) such that it can be cited independently in the future. For instructions see: http://journals.plos.org/plosone/s/submission-guidelines#loc-laboratory-protocols

We look forward to receiving your revised manuscript.

Kind regards,

Igor B. Rogozin

Academic Editor

PLOS ONE

Journal Requirements:

**When submitting your revision, we need you to address these additional requirements:**

**Please ensure that your manuscript meets PLOS ONE's style requirements, including those for file naming. The PLOS ONE style templates can be found at http://www.plosone.org/attachments/PLOSOne_formatting_sample_main_body.pdf and http://www.plosone.org/attachments/PLOSOne_formatting_sample_title_authors_affiliations.pdf** Thank you for stating in your Funding Statement: [The funders had no role in study design, data collection and analysis, decision to publish, or preparation of the manuscript.]. Please provide an amended statement that declares *all* the funding or sources of support (whether external or internal to your organization) received during this study, as detailed online in our guide for authors at http://journals.plos.org/plosone/s/submit-now.  Please also include the statement “There was no additional external funding received for this study.” in your updated Funding Statement.Please include your amended Funding Statement within your cover letter. We will change the online submission form on your behalf.

Additional Editor Comments (if provided):

Dear Dr. Ding,

Please, implement suggestions, I do not see major problems.

Best regards, Igor

Reviewers' comments:

Reviewer's Responses to Questions

**Comments to the Author**

1. Is the manuscript technically sound, and do the data support the conclusions?

Reviewer #1: Yes

Reviewer #2: Yes

2. Has the statistical analysis been performed appropriately and rigorously? 

Reviewer #1: Yes

Reviewer #2: Yes

3. Have the authors made all data underlying the findings in their manuscript fully available?

Reviewer #1: Yes

Reviewer #2: Yes

4. Is the manuscript presented in an intelligible fashion and written in standard English?

Reviewer #1: Yes

Reviewer #2: Yes

5. Review Comments to the Author

Reviewer #1: Ding and colleagues set out to discern whether the direct disruption of long non-coding RNA (lncRNA) could be a mechanism involve in the association of SNPs and rheumatoid arthritis (RA) susceptibility. They assessed the question by evaluating the overlap between SNPs associated with RA susceptibility, lncRNA annotations and enhancer annotations. The authors concluded that the RA susceptibility genetic variants are more likely to act via enhancers than lncRNA. Since the subtle enriched overlap between SNPs associated with RA and lncRNAs are not independent from the strong overlapped enrichment between RA SNPs and enhancers.

This study contributes to comprehend which are the mechanisms behind the statistical associations of SNPs with the risk to develop RA. Importantly, the authors present an improvement on the methodology to assess functional genomic information in relation to GWAS results. This is to condition the analysis with the chromatin state information, in order to avoid cofounders on the enrichment analysis.

I would like to point minor changes which may improve the future reproducibility of the study and a couple of challenges that may or may not be in the scope of the study.

1. Would be possible for the authors to create a graph or a visual interpretation of the analysis pipeline implemented in the study?

2. Please add more details about the RA GWAS summary statistics employ in the analysis. For instance, how many SNPs and max and min p-value of association (~range of p-values). In order to provide the reader relevant information without having to search for the referenced paper.

3. Could you please add more details in the methodology to the “immune-relevant enhancers definition” (page 6 lines 98 to 101)? For example, the flow cytometry antibodies (or markers) reference by the Roadmap Epigenome Project to the “Blood and T cells” or “HSC and B cell”. It could be done adding them to the text or a link to Roadmap Epigenome Project.

4. A similar query as the previous one for the expression profiling section. Please add the specific cell type to which the SRA accession numbers refers to. For instance, is it from adult or fetal cells? Which flow cytometry markers?

5. Could the authors briefly explain in the methods section how to interpret the statistic (i.e. log2 enrichment) they used to “quantify” enrichment? E.g. how does it relate to the two-side Welch’s test or is it the output of the fgwas? This to facilitate the understanding added to the descriptors of the enrichment measure (i.e. negligible in line 159)

Other points

1. Did the authors consider to test the lncRNA/enhancer enrichment in other cellular types known to be relevant in the inflammation site of RA, such fibroblasts?

2. Would it be possible to narrow down the immune-related enhancer enrichment to more specific cell types? In a similar fashion as presented in figure 2b.

3. Unfortunately, the figure 1 was not readable, possibly due to a combination of the original format and the journal submission system. Therefore, my comments are based on the rest of the content of the manuscript.

Reviewer #2: This manuscript reports an analysis of the potential role of genetic variants, identified from genome-wide association studies (GWAS) in rheumatoid arthritis, on mediating their effects through long non-coding RNAs (lncRNAs). The authors use publically available GWAS data, chromatin and RNA-seq data from the Roadmap Epigenomics Project and the fgwas algorithm to undertake these analyses.

The authors report a significant enrichment for immune-enriched lncRNA, but this enrichment was no longer apparent when conditioned on immune-relevant enhancers. The authors therefore suggest that direct disruption of lncRNA sequence, independent of enhancer disruption, does not represent a major mechanism by which susceptibility to complex diseases is mediated.

This paper is clearly written and well executed. I have some generally minor comments; these relate primarily to a more nuanced and more balanced discussion of the data.

Materials and Methods

For clarity and ease of reading, it would be better to summarise the data on the Roadmap Epigenomics 18-state model in a table.

Results

Lines 205-212. The assumption of normality for the two-sided Welch’s t-test to test “Statistical difference between the distributions of immune-enriched lncRNA and mRNA abundance” is clearly not valid (Fig 4). The authors should either use a more appropriate alternative test, or comment on this.

Discussion

I would like to see included some more nuanced and balanced discussion relating to the effects of lncRNAs that are not mediated through direct disruption of DNA sequence. This should include some discussion of the fact that the relatively low lncRNA expression levels may not correlate with effect size /biological relevance of specific lncRNAs. lncRNAs also show higher developmental stage specificity (as well as cell type specificity) than mRNAs, with regard to the data sources used, and the fact that these publically available resources reflect the healthy, rather than disease, state.

Minor comments on syntax

Methods. Enrichment testing. Define ‘TSS’ and ‘HSC’ at first use.

Results, line 186. Fig 3 is duplicated.

Typo fig 4A axis label.

Line 204. Complexes diseases.

6. PLOS authors have the option to publish the peer review history of their article (what does this mean?). If published, this will include your full peer review and any attached files.

Reviewer #1: No

Reviewer #2: No

---

## [Author Response · Author response to Decision Letter 0]

31 Jan 2020

Comments from all reviewers are gratefully received; we believe that by implementing the following recommendations the rigor and ease of understanding of our manuscript has been improved and would like to thank all reviewers for their contributions towards this.

A full list of how each of the reviewers’ comments have been addressed is found below. Please note that line numbers relate to the revised manuscript with track changes.

I hope that you are satisfied with the manner in which these comments have been addressed and agree that they have improved the manuscript. Thank you once again for contributing to our manuscript in such a constructive manner.

Yours sincerely,

Dr James Ding

REVIEWER #1:

1. Would be possible for the authors to create a graph or a visual interpretation of the analysis pipeline implemented in the study?

A visual interpretation of the analysis pipeline has been added as an additional figure (Fig1) and referenced in the text at the end of the introduction. We hope this clarifies the approach taken.

2. Please add more details about the RA GWAS summary statistics employ in the analysis. For instance, how many SNPs and max and min p-value of association (~range of p-values). In order to provide the reader relevant information without having to search for the referenced paper.

We thank the reviewers for this comment as we had neglected to highlight that we restricted our analysis to the European meta-analysis as we believe this corresponds best to the ethnicity of the donors that contribute to other data-sources in our analysis. We apologise for this oversight and have sought to clarify this in the text, such that line 160-162 now reads: “…using a more inclusive approach that incorporates the probability of association for all imputed SNPs taken from the most recent European RA GWAS meta-analysis (8 514 610 SNPs, p value ranges from 1.0 to 1.94 x 10-280) [1]”. We hope that this clarifies that no threshold was used to determine which SNPs contributed to our analysis.

3. Could you please add more details in the methodology to the “immune-relevant enhancers definition” (page 6 lines 98 to 101)? For example, the flow cytometry antibodies (or markers) reference by the Roadmap Epigenome Project to the “Blood and T cells” or “HSC and B cell”. It could be done adding them to the text or a link to Roadmap Epigenome Project.

We have included additional detail in the methods section of our paper to describe how immune-relevant enhancers were defined. This includes reference to the 6 histone marks used to contribute to the Roadmap Epigenomics Projects 18-state model (lines 109-111) and additional description of the cell types contributing to the Roadmap Epigenomics Project “Blood and T cells” or “HSC and B cell” cell type/tissue group. The relevant section (lines 125-130) now reads: “Immune-relevant enhancers were defined as genomic regions annotated as enhancers in cell-types originating from “Blood and T cell” or “HSC and B cell” anatomical locations by the Roadmap Epigenomics project [12]. These include a total of 17 partially overlapping primary cell populations, isolated from peripheral and umbilical cord blood; including peripheral blood mononuclear cells, B cells, Natural killer cells, haematopoietic stem cells, and various subtypes of T cells.”

Additional detail can be found in Figure 2 and Supplementary Table 1 of Kundaje et al. 2015 (referenced in our manuscript as reference 12). In addition, experimental protocols from the roadmap epigenomics website detail that “Hematopoietic cells were provided as a service by the S. Heimfeld Laboratory at the Fred Hutchinson Cancer Research Center, Seattle, WA and include CD3+, CD4+, CD8+, CD14+, CD19+/CD20+, CD34+, and CD56+ cells, from both mobilized and non-mobilized donors. Cells were obtained from human leukapheresis product using standard procedures. Briefly, the lymphocyte subclasses were isolated by immunomagnetic separation using the CliniMACS affinity-based technology (Miltenyi Biotec GmbH, Bergisch Gladbach, Germany) according to the manufacturer’s recommendations.”

4. A similar query as the previous one for the expression profiling section. Please add the specific cell type to which the SRA accession numbers refers to. For instance, is it from adult or fetal cells? Which flow cytometry markers?

We have included additional detail taken from the National Centre for Biotechnology Information’s database, corresponding to the age, gender and ethnicity of these samples (lines 137-139). Unfortunately it has not been possible to include flow-cytometry/immunomagnetic markers as these are not clearly reported by the Roadmap Epigenomics Project. In Supplementary Table 1 of Kundaje et al. 2015 these cells are referred to as CD4+ and CD25- “Primary T helper cells from peripheral blood”.

5. Could the authors briefly explain in the methods section how to interpret the statistic (i.e. log2 enrichment) they used to “quantify” enrichment? E.g. how does it relate to the two-side Welch’s test or is it the output of the fgwas? This to facilitate the understanding added to the descriptors of the enrichment measure (i.e. negligible in line 159)

This has been clarified, through the inclusion of an additional sentence in the methods section (line 104-107): “Enrichment estimates are reported as outputted by fgwas, on a log2 scale, such that positive values indicate an enrichment and negative values indicate a depletion of the given annotation amongst disease susceptibility variants.”

Other points

1. Did the authors consider to test the lncRNA/enhancer enrichment in other cellular types known to be relevant in the inflammation site of RA, such fibroblasts?

We agree that it would be of benefit to the manuscript to include additional disease relevant cell types, such as synovial fibroblasts, or fibroblast-like synoviocytes (FLS), however data sufficient to perform a comparable analysis with that performed for other cell types is not publically available. FLS show many characteristics common with fibroblasts, but they also secrete unique proteins, that are normally absent in other fibroblast lineages. Nevertheless, two primary foreskin fibroblast cell types are included in what is now Fig2, corresponding to Roadmap Epigenomics epigenome IDs E055 and E056. An enrichment of enhancer annotations from these samples was not observed.

2. Would it be possible to narrow down the immune-related enhancer enrichment to more specific cell types? In a similar fashion as presented in figure 2b.

This has effectively been performed in what is now Fig2, which shows the extent to which enhancers annotated in all Roadmap Epigenomics epigenomes are enriched amongst RA susceptibility variants. Whilst there is some variability in the extent of this enrichment, and we report the cell type which gives the highest enrichment estimate (regulatory T cells, line 174), confidence intervals overlap to the extent that no firm comparisons can be drawn.

3. Unfortunately, the figure 1 was not readable, possibly due to a combination of the original format and the journal submission system. Therefore, my comments are based on the rest of the content of the manuscript. 

This is regrettable, especially since it appears to have relevance to the two previous points. We apologise for any contribution we have made to this and would encourage this reviewer to review this figure on bioRxiv, should it remain unreadable upon submission of our revised manuscript (https://doi.org/10.1101/804880).

REVIEWER #2:

1. Materials and Methods: For clarity and ease of reading, it would be better to summarise the data on the Roadmap Epigenomics 18-state model in a table.

This information has been transferred to a table, which we agree aids the clarity and ease of reading.

2. Results: Lines 205-212. The assumption of normality for the two-sided Welch’s t-test to test “Statistical difference between the distributions of immune-enriched lncRNA and mRNA abundance” is clearly not valid (Fig 4). The authors should either use a more appropriate alternative test, or comment on this.

Thank you for this comment, we had struggled to calculate the statistical significance of the difference between the distributions shown in what is now Fig 5, but have now found a more suitable test. Due to the large sample sizes and the central limit theorem the Z-score can be assumed to approximate a normal distribution. The significance of both comparisons is now reported as “p = 1.04 x 10-34” and “p = 1.01 x 10-24” in the results section, with the methods section (lines 147-149) reading: “Statistical difference between the distributions of immune-enriched lncRNA and mRNA abundance was established using a two-sided Z-test, without assuming equal variance.”

3. Discussion: I would like to see included some more nuanced and balanced discussion relating to the effects of lncRNAs that are not mediated through direct disruption of DNA sequence. This should include some discussion of the fact that the relatively low lncRNA expression levels may not correlate with effect size /biological relevance of specific lncRNAs. lncRNAs also show higher developmental stage specificity (as well as cell type specificity) than mRNAs, with regard to the data sources used, and the fact that these publically available resources reflect the healthy, rather than disease, state.

We have expanded our discussion (lines 289-307) in order to better cover lncRNA that are either not captured within the data sources used or function independent of their underlying DNA sequence. This includes highlighting the potential disparity between the extent to which a lncRNA may be expressed and its biological significance.

Minor comments on syntax

1. Methods. Enrichment testing. Define ‘TSS’ and ‘HSC’ at first use.

2. Results, line 186. Fig 3 is duplicated.

3. Typo fig 4A axis label.

4. Line 204. Complexes diseases.

These minor comments have all been acted upon with the underlying errors having been corrected. Thank you for your careful revision.

---

## [Editor Report · Decision Letter 1]

20 Feb 2020

Exploring the overlap between rheumatoid arthritis susceptibility loci and long non-coding RNA annotations

PONE-D-19-27368R1

Dear Dr. Ding,

We are pleased to inform you that your manuscript has been judged scientifically suitable for publication and will be formally accepted for publication once it complies with all outstanding technical requirements.

With kind regards,

Igor B. Rogozin

Academic Editor

PLOS ONE

Additional Editor Comments (optional):

The authors implemented suggestions, the paper is acceptable.
---

## [Editor Report · Acceptance letter]

4 Mar 2020

PONE-D-19-27368R1 

Exploring the overlap between rheumatoid arthritis susceptibility loci and long non-coding RNA annotations 

Dear Dr. Ding:

I am pleased to inform you that your manuscript has been deemed suitable for publication in PLOS ONE. Congratulations! Your manuscript is now with our production department. 

With kind regards,

on behalf of

Dr. Igor B. Rogozin 

Academic Editor

PLOS ONE